# Addition is almost all you need: Compressing large language models with double binary factorization

**Vladimír Boža**                                                                      *boza@fmph.uniba.sk*
*Faculty of Mathematics, Physics and Informatics*
*Comenius University, Bratislava*

**Vladimír Macko**                                                       *vladimir.macko@fmph.uniba.sk*
*Faculty of Mathematics, Physics and Informatics*
*Comenius University, Bratislava*

**Reviewed on OpenReview:** *https://openreview.net/forum?id=k5kUKoewdQ*

## Abstract

Binary quantization approaches, which replace weight matrices with binary matrices and substitute costly multiplications with cheaper additions, offer a computationally efficient approach to address the increasing computational and storage requirements of Large Language Models (LLMs). However, the severe quantization constraint ($\pm 1$) can lead to significant accuracy degradation. In this paper, we propose Double Binary Factorization (DBF), a novel method that factorizes dense weight matrices into products of two binary (sign) matrices, each accompanied by scaling vectors. DBF preserves the efficiency advantages of binary representations while achieving compression rates that are competitive with or superior to state-of-the-art methods. Specifically, in a 1-bit per weight range, DBF is better than existing binarization approaches. In a 2-bit per weight range, DBF is competitive with the best quantization methods like QuIP# and QTIP. Unlike most existing compression techniques, which offer limited compression level choices, DBF allows fine-grained control over compression ratios by adjusting the factorization's intermediate dimension. Based on this advantage, we further introduce an algorithm for estimating non-uniform layer-wise compression ratios for DBF, based on previously developed channel pruning criteria.

The code is available at: `https://github.com/usamec/double_binary`

## 1 Introduction

Large language models (LLMs) have achieved unprecedented success in various natural language processing and reasoning tasks. However, the increasing scale of these models has led to substantial computational and storage demands, posing significant challenges for deployment. To address these limitations, various compression techniques such as quantization (Frantar et al., 2022; Malinovskii et al., 2024; Tseng et al., 2024a;b), and pruning (Frantar & Alistarh, 2023; Boža, 2024) have reemerged, aiming to reduce model size and inference latency without significant loss in performance. Moreover, methods like BitNet (Wang et al., 2023) and OneBit (Xu et al., 2024) restrict weight matrices to binary values and replace energy-costly multiplication with more energy-efficient addition.

In this paper, we push weight binarization further. We propose to replace each weight matrix with a product of two binary sign matrices, each scaled by appropriate vectors. We present a heuristic algorithm for calculating such factorization and show that it results in superior compression compared to a single sign matrix and is competitive with the state-of-the-art quantization approaches.

**Summary of contributions.** We propose a practical algorithm for factorizing dense weight matrices into a product of two binary matrices (with appropriate scaling factors). We apply this factorization to

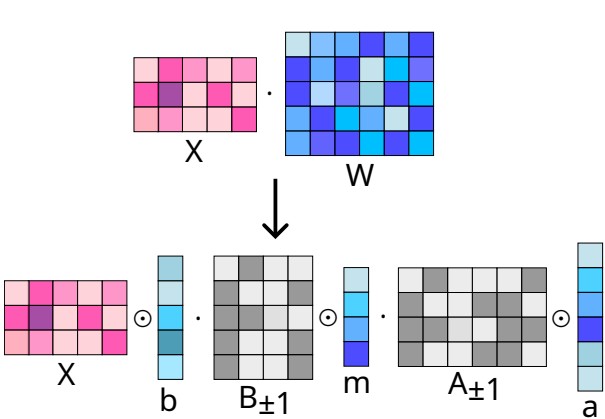 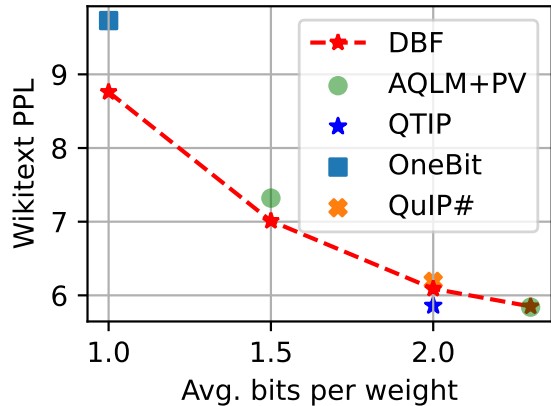

Figure 1: Left: Schematic drawing of computation in Double binary factorization. $X$ is the layer input in FP16 format, $W$ is the original weight matrix. $A$ and $B$ consist of $\pm 1$ elements, $a, m, b$ are vectors in FP16 format. We omit matrix transpositions for simplicity. Right: Comparison of our Double binary factorization (DBF) with previously proposed network compression methods on Llama2-7B.

LLM compression and achieve results better than single-matrix binarization and competitive with leading quantization methods. Our factorization is practical for deployment on current GPUs, achieving 2-3.5x speedups during inference. Furthermore, since multiplication with binary matrices requires only additions, our approach has potential for significant energy savings.

DBF also has multiple notable advantages compared to other weight compression methods. First of all, it can take weight importance into account and gives lower error to weights with higher importance. Furthermore, DBF can also achieve a fine-grained range of compression ratios by varying the size of the middle dimension of the factorization. Typical compression approaches allow only limited choices of compression ratios and are commonly limited to an integer number of bits per weight. Finally, we propose an iterative algorithm to assign different compression ratios to each layer. We show that we can treat the middle dimension of factorization as channels and use previously devised channel pruning criteria to iteratively prune the middle dimension of factorization.

## 2 Related work

**Post-Training quantization (PTQ).** There are many methods for post-training quantization of LLMs. Almost all of them work by solving some variation of the layer-wise compression problem. GPTQ (Frantar et al., 2022) improves scalar quantization by minimizing layer-wise errors. QuIP# (Tseng et al., 2024a) uses incoherence preprocessing and lattice codebooks to further improve quantization accuracy. AQLM (Egiazarian et al., 2024) uses learned vector codebooks. QTIP (Tseng et al., 2024b) combines incoherence preprocessing with trellis coding to get even better accuracy than previous methods. However, a key drawback of these more sophisticated quantization methods is that they require decompressing weights back to full precision and performing full-precision multiplications, thus preventing them from utilizing hardware acceleration optimized for lower-precision arithmetic. Additionally, most existing methods support only a limited set of compression ratios (e.g., QTIP needs an integer number of bits per weight), reducing their flexibility.

**Models with binary weights.** Networks with binary weights (Rastegari et al., 2016) were always attractive, due to the potential of simple computation, where one replaces multiplication with additions. This idea was applied to LLMs in BitNet (Wang et al., 2023), which showed a possibility of training LLMs with binary weights. OneBit (Xu et al., 2024) showed a simple way of converting each weight matrix into a binary

matrix during post-training quantization. BiLLM (Huang et al., 2024) also binarizes weight matrices, but uses another binary matrix to enhance precision for a selected subset of important parameters.

**Weight factorization.** Rather than quantizing weight matrices, another approach is to approximate them as a product of several smaller matrices. The most common method is low-rank factorization (Li & Shi, 2018; Jaderberg et al., 2014), where an $n \times m$ matrix is decomposed into the product of an $n \times k$ and $k \times m$ matrix, with $k \ll \min(n, m)$. However, low-rank factorizations come with severe degradation in accuracy. Boža & Macko (2025) showed that one can factorize the weight matrix into two sparse matrices and get better results than using just one sparse matrix with the same amount of nonzeros. Saha et al. (2024) proposed Caldera, which decomposes each weight matrix as a sum of high-precision low-rank factorization and a low-precision quantized matrix. SVD-LLM & SVD-LLM v2 (Wang et al., 2024; 2025) improve over naive SVD by incorporating truncation-aware data whitening to align singular value truncation with compression loss and assigning specific compression ratios based on the varying sensitivity of different layers. Cherniuk et al. (2024) factories convolution layers in vision models into product of multiple quantized matrices. Compared to our work, they use a single scaling factor for the whole tensor and much higher precision (4-bit and higher).

**Finetuning of quantized models.** State-of-the-art LLM compression methods use fine-tuning during and after quantization. Multiple methods fine-tune only the remaining continuous parameters. AQLM (Egiazarian et al., 2024) only finetunes continuous parameters after the quantization. QuIP# (Tseng et al., 2024a) and QTIP (Tseng et al., 2024b) use lightweight fine-tuning during the quantization process and then proceed to fine-tune continuous parameters after quantization.

In most cases, the fine-tuning of discrete parameters of quantized models is done via straight-through estimation (Courbariaux et al., 2014; Jacob et al., 2018). The other option is to use stochastic rounding (Alistarh et al., 2017). Malinovskii et al. (2024) proposed PV-tuning, where they showed that it is better to tune discrete parameters with a higher learning rate, but only tune a small subset of discrete parameters in each update step.

## 3 Double binary factorization

We work in the context of layer-wise compression, where each linear layer in the model is replaced by a compressed version. Our goal is to keep weight matrices in a binary format while having regular 16-bit floating-point activations.

**OneBit decomposition.** Xu et al. (2024) proposed to approximate weight matrices by:

$$W \approx a \odot W_{\pm 1} \odot b^T$$

where $a, b$ are column vectors, $W_{\pm 1}$ is a sign matrix with entries from $\{-1, 1\}$ and $\odot$ denotes elementwise (Hadamard) product with appropriate broadcasting. During linear layer computation, we calculate:

$$XW^T \approx ((X \odot b^T)W_{\pm 1}^T) \odot a^T$$

OneBit decomposition is calculated via Sign-Value-Independent Decomposition (SVID), which first sets $W_{\pm 1} = \texttt{Sign}(W)$ and $ab^T$ is the rank-1 approximation of $|W|$ calculated via SVD or NMF.

**Double binary factorization.** Inspired by Double Sparse Factorization (Boža & Macko, 2025), we propose to factorize the weight matrix into a product of two binary matrices with appropriate vector scaling in between (see also Fig. 1):

$$W \approx (a \odot A_{\pm 1} \odot m^T)(B_{\pm 1} \odot b^T) = (a \odot A_{\pm 1})(m \odot B_{\pm 1} \odot b^T)$$

where $a, m, b$ are vectors and $A_{\pm 1}, B_{\pm 1}$ are sign matrices, which translates into the following computation during network forward pass:

$$XW^T \approx ((((X \odot b^T)B_{\pm 1}^T) \odot m^T)A_{\pm 1}^T) \odot a^T$$

**Middle dimension size.** The original matrix $W$ has a shape $n \times m$. Vectors $a$ and $b$ must have sizes $n$ and $m$ respectively. But we can select the middle dimension size $k$ to match the desired compression ratio. Then the vector $m$ will have size $k$, and matrices $A_{\pm 1}$ and $B_{\pm 1}$ will have size $n \times k$ and $k \times m$.

For example, if the input matrix is squared and we target approximately 1-bit compression, we will select $k = n/2$. If we target ~2-bit compression, we will select $k = n$. In general, we will select $k = b\frac{nm}{n+m}$, where $b$ is the desired average number of bits per original weight.

### 3.1 Practical considerations for double binary factorization

**Storage size.** By controlling the middle dimension of factorization, we can compress the matrix to any desirable size. This is a huge advantage compared to many other methods, which support a limited number of sizes (e.g., scalar quantization supports only integer values as a number of bits per weight). We also need to store scaling vectors, but their size is negligible (e.g., when compressing a matrix of size $4096 \times 4096$ to 2 bits/weight, storing scaling vectors costs 0.012 bits per weight).

**Inference costs.** It is true that the total number of operations for DBF is higher than for ordinary scalar quantization (except for the case where the number of bits per weight is less than or equal to 1), but in all cases, we replace costly multiplications with much simpler additions. Moreover, LLM inference is often bound by memory transfer, and memory transfer costs for DBF are similar to the costs of any other quantization with an equal number of bits per weight. We measure actual matrix multiplication timings in the experimental section and found that DBF is 2-3.5x faster than dense baseline when using 2 bits per weight. Moreover, Wang et al. (2023) showed that using binary weights can save energy during inference since one needs only additions instead of multiplications. This could lead to even better savings in the future with proper HW support.

**Storage of middle activations during finetuning.** As with any other weight factorization, one needs to store the middle activations during fine-tuning, which incurs nontrivial memory costs. Similar to Double sparse factorization (Boža & Macko, 2025), we found that when using gradient checkpointing, this storage increase is practically negligible.

### 3.2 Computing Double Binary Factorization

Computing optimal Double binary factorization is probably an NP-hard problem, but we respond to that with "Here's where the fun begins." (Solo, 1977) and propose a heuristic algorithm for computing DBF. We compute Double Binary Factorization very similarly to Double Sparse Factorization (DSF) (Boža & Macko, 2025). Our main goal is to minimize:

$$\min ||W - (a \odot A_{\pm 1} \odot m^T)(B_{\pm 1} \odot b^T)||_2^2$$

First, we split the middle scaling factor into two (we can easily put it back by multiplying the middle factors together):

$$\min ||W - (a \odot A_{\pm 1} \odot m_1^T)(m_2 \odot B_{\pm 1} \odot b^T)||_2^2$$

Denote $A = a \odot A_{\pm 1} \odot m_1^T$ and $B = m_2 \odot B_{\pm 1} \odot b^T$. We run alternating minimization, where we first initialize $A$ with a random matrix, then fix $A$ and optimize $B$, then fix $B$ and optimize $A$, etc.

The main subproblem of the alternating minimization is:

$$\min_{A} \quad ||AB - W||_F$$
$$\text{s. t.} \quad A = a \odot A_{\pm 1} \odot m_1^T$$

We solve this problem using the Alternating direction method of multipliers (ADMM) (Boyd et al., 2011). ADMM is an iterative algorithm for solving constrained optimization problems. While ADMM provably

converges only for convex problems, and our constraint is non-convex, there have been numerous successful attempts to use ADMM for non-convex problems.

We use ADMM for solving constrained optimization of the form:

$$\text{minimize} \quad f(x)$$
$$\text{subject to} \quad x \in C$$

In this case, the one iteration of ADMM is ($\rho$ is a penalty factor, usually set to one in our case, and $u$ are scaled dual variables):

$$x^{k+1} = \arg\min_x f(x) + (\rho/2)||x - z^k + u^k||_2^2$$
$$z^{k+1} = \Pi_C(x^{k+1} + u^k)$$
$$u^{k+1} = u^k + x^{k+1} - z^{k+1}$$

Where $\Pi_C$ is a Euclidean projection onto the set $C$. In our case, we want to project onto the set of matrices, which can be factorized as $a \odot A_s \odot m_1^T$. We will use SVID projection from OneBit (Xu et al., 2024), i.e., we will calculate $\texttt{SVID}(Z)$ as follows: First, we set $Z_{\pm 1} = \texttt{Sign}(Z)$, and do rank-1 decomposition of $|Z|$ into $am_1^T$. Then $\texttt{SVID}(Z) = a \odot Z_{\pm 1} \odot m_1^T$. Since we need to do a lot of iterations, we compute the rank-1 decomposition using power iteration.

Then the one ADMM update becomes:

$$\widehat{W}^{(k+1)} = (B^T B + \rho I)^{-1}(B^T W + \rho(A^{(k)} - U^{(k)}))$$
$$A^{(k+1)} = \texttt{SVID}(\widehat{W}^{(k+1)} + U^{(k)})$$
$$U^{(k+1)} = U^k + \widehat{W}^{(k+1)} - A^{(k+1)}$$

We also use all heuristic improvements from DSF (warm-starting of inner iterations). We also normalize rows of matrix $B$; we prefer to use fewer inner updates (ADMM steps) and more outer updates (alternating minimization steps), and we reuse solutions from previous inner iterations.

### 3.3 Input and output importance scaling

While many compression schemes try to solve the layer-wise pruning problem ($\min ||XW - XW_c||_2^2$, where $X$ is the calibration input, $W$ the original matrix, and $W_c$ the compressed matrix) (Frantar & Alistarh, 2023; Frantar et al., 2022; Tseng et al., 2024a), we opt to use a slightly different approach.

We assign different importance to preserving the rows and columns of the matrix $W$. We use input activation norm (similar to Wanda (Sun et al., 2023)) as the column (input) importance. We use the gradient norm as row (output) importance. Both input activation norms and gradient norms are precalculated using a small calibration dataset. One can think about scaling by activation norm and gradient norm as a crude rank-1 approximation to the diagonal Fisher matrix.

We can easily incorporate input and output importance into our factorization algorithm. We first calculate $W' = o \odot W \odot i^T$, where $o$ are gradient norms and $i$ are input activation norms. We then factorize $W' \approx (a' \odot A_{\pm 1} \odot m^T)(B_{\pm 1} \odot b'^T)$ and scale scaling vectors back: $a = a'/o$, $b = b'/i$.

### 3.4 Fine-tuning during and after factorization

Almost all state-of-the-art quantization algorithms use some form of fine-tuning. We use a slight variation of the procedure from QuIP# (Tseng et al., 2024a) and QTIP (Tseng et al., 2024b) as follows: We use a small calibration dataset (in our case, 256 sequences). When compressing the $i$-th transformer block, we collect its expected output $Y^{(i)}$ from the original dense model. We also collect the output of the $(i-1)$-th block

after compression, which becomes the input $X^{(i)}$ of the $i$-th block. Before compression, we first fine-tune the block to correct errors from previous blocks. Then we compress the query, value, and output matrices, fine-tune the rest of the block, and compress the remaining matrices. After the compression, we fine-tune the remaining unquantized parameters (scaling vectors in each binary factorization).

**PV-tuning.** We also attempt to fine-tune signs in the binary matrices. This is challenging since storing dense weights (e.g., for straight-through estimation or PV tuning) requires more memory than the dense model. We reduce the memory requirements by always running PV-tuning only on a subset of layers. In each $k$-step (we set $k = 50$), we select a random subset of layers to be PV-tuned, where each layer has a probability of 1/10 of being selected. We also fine-tune all continuous parameters in all layers.

### 3.5 Nonuniform layer compression ratios

Most quantization algorithms quantize all layers to the same bitwidth. In many cases, they offer limited compression sizes (i.e. integer avg. bits per layer) and combining different ones is non-trivial.

Evopress (Sieberling et al., 2024) proposes to use evolutionary algorithms to select layer compression ratios, while HIGGS (Malinovskii et al., 2025) shows that for lower compression ratios, one can find a linear relationship between layer-wise compression errors and final model perplexity. While both approaches select good compression ratios, we found that the improvement disappears after fine-tuning.

We propose a different approach based on two observations about DBF. First of all, DBF can have almost any compression ratio (even constraining the middle dimension to multiples of 32 leads to 0.03 bits/weight steps in compression ratio or smaller). Secondly, after we compute DBF for a higher compression ratio, we can consider the middle dimension as channels and use the previously proposed channel pruning methods. Specifically, we use the criteria used in Yang et al. (2023); Molchanov et al. (2019). Considering our factorization $(a \odot A_{\pm 1} \odot m^T)(B_{\pm 1} \odot b^T)$ for each $m_i$ we calculate score $s_i$ as (we sum over multiple batches):

$$s_i = \sum_b \left( \frac{\partial E}{\partial m_i} m_i \right)^2$$

We then sort scores $s_i$ in all layers with the same size (e.g., in Llama3 we group (k,v), (o,q), (up,gate,down) layers together) and keep only middle channels with the highest scores. This gives us the desired sizes of each layer, and we run the compression process again. This naturally gives an iterative compression process.

## 4 Experiments

### 4.1 Uniform compression of LLMs

We evaluate our proposed Double binary factorization on compressing Llama2-7B (Touvron et al., 2023) and Llama3-8B (Grattafiori et al., 2024) models.

**Calibration and fine-tuning data.** We use Redpajama dataset (Weber et al., 2024) for calibration and fine-tuning. We use the same preprocessing as in Malinovskii et al. (2024). We select 256 random sequences as a calibration dataset and use them to compute input activation norms and output gradient norms for each linear layer. We also use it for fine-tuning during compression.

**Compute resources.** Compression with fine-tuning during compression was done on one RTX 4090 GPU and takes 6-8 hours. We further run fine-tuning on 4 A100 GPUs for 24 hours.

**Evaluation metrics.** We evaluate compression methods using WikiText-2 (Merity et al., 2016) perplexity and zero-shot accuracy on ARC-easy and ARC-challenge (Clark et al., 2018), PiQA (Bisk et al., 2020) and Winogrande (Sakaguchi et al., 2021). We also evaluate Llama3-8B 2-2.3 bit models on MMLU 5-shot (Hendrycks et al., 2020) and GSM8k 8-shot (Cobbe et al., 2021) benchmarks.

**Compared methods.** We compare against multiple state-of-the-art methods in 1-2.3 bit compression range. We compare with AQLM (Egiazarian et al., 2024) with PV-tuning (Malinovskii et al., 2024). We only

Table 1: Results for uniform compression of Llama2-7B. Avg. bits refers to the average number of bits per original weight.

| Avg. bits | Method | WikiText ppl. | ArcC | ArcE | PiQA | Wino | Avg. zero shot |
|---|---|---|---|---|---|---|---|
| 16 | Dense | 5.12 | 43.43 | 76.34 | 78.07 | 69.06 | 66.73 |
| 2.3 | AQLM + PV | 5.84 | 38.91 | 72.90 | 77.37 | 67.72 | 64.23 |
| 2.4 | Caldera | 5.84 | 35.90 | 64.60 | 76.50 | 65.70 | 60.68 |
| 2.3 | DBF | 5.87 | 38.39 | 73.35 | 76.49 | 66.85 | 63.77 |
| 2.3 | DBF + PV | 5.85 | 39.33 | 73.65 | 76.55 | 67.08 | 64.15 |
| 2 | QTIP | 5.86 | 39.76 | 73.31 | 76.27 | 67.20 | 64.14 |
| 2 | QUIP# | 6.19 | 37.79 | 71.88 | 75.46 | 65.66 | 62.70 |
| 2.1 | Caldera | 6.30 | 35.40 | 63.30 | 75.40 | 64.60 | 59.68 |
| 2 | DBF | 6.14 | 37.03 | 72.05 | 76.22 | 65.82 | 62.78 |
| 2 | DBF + PV | 6.09 | 36.94 | 71.88 | 76.16 | 66.06 | 62.76 |
| 1.5 | AQLM + PV | 7.32 | 29.44 | 64.14 | 73.12 | 63.38 | 57.52 |
| 1.5 | DBF | 7.16 | 35.06 | 66.32 | 73.28 | 64.40 | 59.77 |
| 1.5 | DBF + PV | 7.01 | 35.58 | 67.12 | 73.72 | 63.69 | 60.03 |
| 1 | OneBit | 9.73 | 29.61 | 41.58 | 68.12 | 58.41 | 49.43 |
| 1.1 | BiLLM | 32.48 | 24.40 | 36.20 | 60.60 | 52.40 | 43.40 |
| 1 | DBF | 9.57 | 26.45 | 55.30 | 67.41 | 58.87 | 52.01 |
| 1 | DBF + PV | 8.76 | 27.47 | 57.87 | 68.66 | 58.40 | 53.10 |

Table 2: Results for uniform compression of Llama3-8B. Avg. bits refer to the average number of bits per original weight.

| Avg. bits | Method | WikiText ppl | ArcC | ArcE | PiQA | Wino | Avg. zero shot |
|---|---|---|---|---|---|---|---|
| 16 | Dense | 5.54 | 50.43 | 80.09 | 79.71 | 72.61 | 70.71 |
| 2.3 | AQLM + PV | 6.76 | 42.32 | 75.46 | 78.45 | 71.67 | 66.98 |
| 2.4 | Caldera | 7.34 | 42.30 | 73.60 | 76.50 | 70.30 | 65.68 |
| 2.3 | DBF | 6.97 | 45.73 | 76.93 | 77.47 | 70.32 | 67.61 |
| 2.3 | DBF + PV | 6.86 | 45.56 | 76.76 | 78.01 | 71.90 | 68.06 |
| 2 | QTIP | 7.33 | 44.20 | 75.20 | 77.60 | 70.70 | 66.93 |
| 2 | QUIP# | 7.84 | 39.20 | 72.90 | 75.60 | 68.20 | 63.98 |
| 2.1 | Caldera | 8.06 | 40.00 | 71.50 | 76.00 | 69.50 | 64.25 |
| 2 | DBF | 7.48 | 41.04 | 74.45 | 76.82 | 67.95 | 65.07 |
| 2 | DBF + PV | 7.30 | 44.45 | 75.33 | 76.98 | 70.08 | 66.71 |
| 1.5 | AQLM + PV | 9.43 | 32.68 | 65.78 | 72.63 | 64.40 | 58.87 |
| 1.5 | DBF | 9.37 | 33.61 | 67.59 | 73.99 | 64.64 | 59.96 |
| 1.5 | DBF + PV | 9.05 | 35.58 | 68.68 | 74.10 | 65.43 | 60.95 |
| 1.1 | BiLLM | 28.80 | 17.70 | 36.00 | 56.10 | 51.00 | 40.20 |
| 1 | DBF | 15.61 | 21.67 | 50.25 | 65.17 | 56.43 | 48.38 |
| 1 | DBF + PV | 13.57 | 25.08 | 56.22 | 67.46 | 58.80 | 51.89 |

compare against PV-tuned models, which are publicly available. We also compare against QuIP# (Tseng et al., 2024a), QTIP (Tseng et al., 2024b) and Caldera (Saha et al., 2024). Finally, we compare against OneBit (Xu et al., 2024) and BiLLM (Huang et al., 2024), which are specifically designed for extreme 1-bit quantization. For our method, we always report numbers with simple tuning (only continuous parameters) and with PV-tuning.

**Results.** Results for uniform compression are shown in Tab. 1, and 2. Our results are comparable to AQLM with PV-tuning for 2.3 bit compression. We are marginally worse than QTIP but better than QuIP# for 2 bit compression. Both methods require decompression into full precision weight, while DBF uses mostly

Table 3: Results for uniform compression of Llama3-8B on MMLU and GSM8k.

| Avg. bits | Method | WikiText ppl | MMLU | GSM8k |
|---:|---|---:|---:|---:|
| 16 | Dense | 5.54 | 65.29 | 54.89 |
| 2.3 | AQLM + PV | 6.76 | 56.20 | 35.25 |
| 2.3 | DBF + PV | 6.86 | 56.37 | 27.59 |
| 2 | QTIP | 7.33 | 56.20 | 24.79 |
| 2 | DBF + PV | 7.30 | 52.67 | 26.15 |

additions. We are better than Caldera for everything except Llama2-7B compression in 2.3 bit range. For 1-1.5 bit compression, our method is superior to other tested methods. Note that OneBit was fine-tuned for much longer than DBF (OneBit reports 7 days on 8 H100s, we tune for 1 day on 4 A100s), and DBF with only continuous parameters fine-tuning is still better. Also, note that to achieve the average 1 bit per layer, we need the middle dimension of factorization to be smaller than the dimensions of the original matrix. But even with the low-rank bottleneck, DBF is still better than OneBit.

**Results on MMLU and GSM8k.** We evaluated the best-performing models for 2-2.3 bit compression of Llama3-8B. Results are in Tab. 3. Results are slightly erratic. DBF is comparable to AQLM on MMLU, but much worse in GSM8k. On the other hand, QTIP performs very well on MMLU but worse on GSM8k. Note that these are raw pretrained models, which were not specifically tuned for reasoning.

## 4.2 Non-uniform compression of LLMs

Due to a lack of computational resources, we run only one round of iterative pruning. We compress Llama3-8B in the same way as above. We start with DBF with 2.1 bits/weight. We then estimate scores for middle elements of the factorization and compute sizes of each layer. We found that restricting each layer to have at least 1.5 bits/weight leads to slightly better results. We then repeat the compression and fine-tuning process from the start. We are able to push perplexity for Llama3-8B further down from 7.30 to 7.26. We see that a nonuniform distribution gives benefit even with one iteration of redistribution.

## 4.3 Properties of DBF

**Adherence to weight importance.** In this experiment, we test whether DBF has lower approximation error for weights with higher importance. We factorize layer 7.self_attn.k_proj from Llama3-8B with input and output importance calculated from the calibration set as in the main experiments. As controls, we use basic 3-bit scalar quantization and OneBit. In case of OneBit, we use importance scaling as described in Sec. 3.3. We plot the results in Fig. 2. We see that neither simple scalar quantization (this is expected) nor OneBit can follow weight importance, while DBF decreases error as the weight importance increases.

**Approximation error vs avg. number of bits.** We also explore how the approximation error changes with the change in compression ratio (avg. number of bits per layer). Again, we compare DBF with scalar quantization and OneBit. This time, we do not use weight importance. We test different compression ratios for DBF and scalar quantization. Results are shown in Fig. 3. We see that for 1-3 bit compression, our method is better. However, at 4 bits and higher, scalar quantization is better than DBF. We found out that we can improve higher bit compression (larger middle sizes) by using "size annealing". We run 80% of iterations with a 2-bit compression. And during the last 20% of iterations, we gradually expand the middle dimension (initializing the expanded part with small random parameters). This slightly improves 3,4 and 6-bit compression and suggest that limited performance for higher bit width is algorithmic problem not fundamental limitation of DBF.

**Scaling to bigger matrices.** We investigate scaling DBF to larger matrices present in even larger language models. We track relative apx. error for different matrices present in Llama3-8B, Llama3-70B and Llama3.1-405B models. Results are shown in Fig. 3. We do not see any degradation for larger matrices.

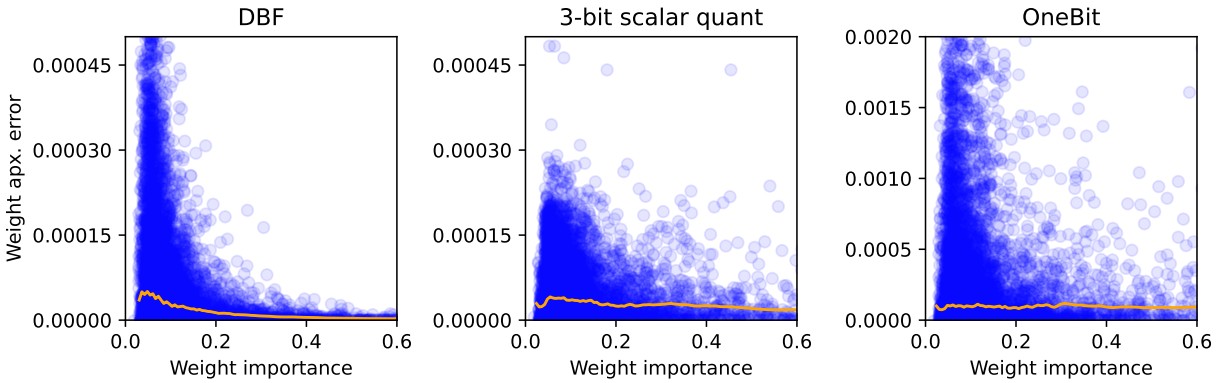

Figure 2: Comparison of weight importance (calculated as a product of input and output importance) vs. approximation error on 7.self_attn.k_proj of Llama3-8B. Blue points are actual matrix elements, and the orange line is a smoothed version.

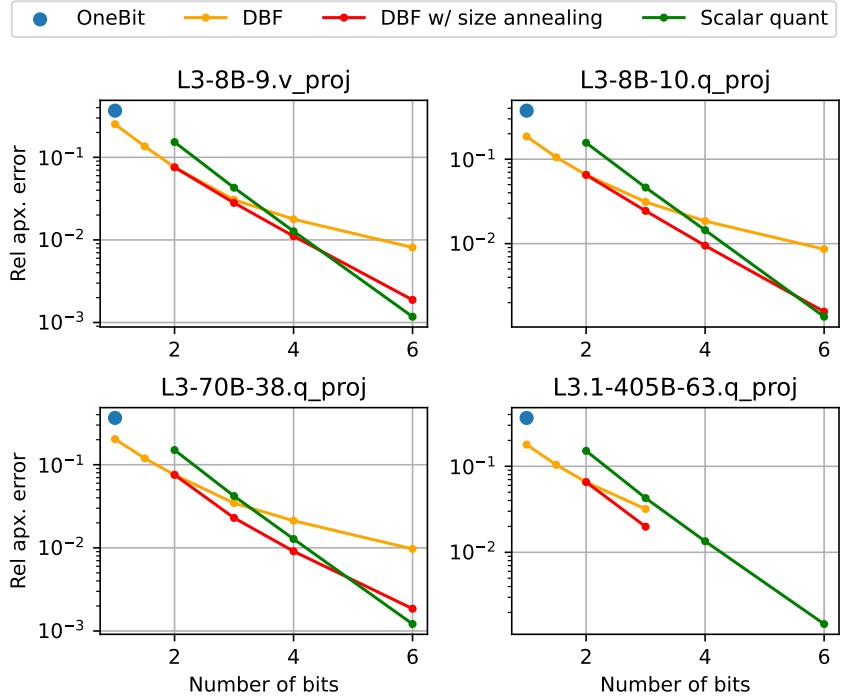

Figure 3: Comparison of number of bits vs. approximation error on two layers of Llama3-8B and on selected q_proj layers of bigger Llama3 models. Compression to 4 and higher number of bits threw OOM error on 40GB GPU for 405B Llama.

Table 4: Comparison of speed of the FP16 matrix-vector multiplication in PyTorch vs DBF for various matrix sizes on Nvidia RTX 4090.

|  | Avg. bits | $4096 \times 4096$ | $4096 \times 14336$ | $8192 \times 8192$ | $8192 \times 28672$ |
|---|---|---|---|---|---|
| Original | 16 | 61 $\mu$s | 153 $\mu$s | 200 $\mu$s | 548 $\mu$s |
| DBF | 2.3 | 29 $\mu$s (x2.14) | 56 $\mu$s (x2.71) | 64 $\mu$s (x3.11) | 167 $\mu$s (x3.25) |
| DBF | 2 | 27 $\mu$s (x2.31) | 51 $\mu$s (x2.98) | 58 $\mu$s (x3.44) | 149 $\mu$s (x3.65) |
| DBF | 1.5 | 24 $\mu$s (x2.61) | 43 $\mu$s (x3.57) | 51 $\mu$s (x3.93) | 115 $\mu$s (x4.71) |
| DBF | 1 | 20 $\mu$s (x3.01) | 37 $\mu$s (x4.17) | 38 $\mu$s (x5.31) | 84 $\mu$s (x6.52) |

Table 5: Batch size 1 decoding throughput on Nvidia RTX 4090 for two versions of Llama models.

|  | Avg. bits | 2-7B | 3-8b |
|---|---|---|---|
| Original | 16 | 68 tok/s | 60 tok/s |
| DBF | 2.3 | 144 tok/s (x2.12) | 121 tok/s (x2.01) |
| DBF | 2 | 153 tok/s (x2.25) | 133 tok/s (x2.22) |
| DBF | 1.5 | 167 tok/s (x2.46) | 153 tok/s (x2.55) |
| DBF | 1 | 170 tok/s (x2.50) | 174 tok/s (x2.90) |

### 4.4 Inference speed

Although our main objective is to show that one can achieve competitive network compression using just binary matrices, we also obtain practical speedups on current hardware. We benchmark matrix-vector multiplication and decoding speeds of LLMs with batch size 1. To perform the computation, we use the implementation of binary matrix multiplication from the package gemlite (Badri & Shaji, 2023).

We report results of matrix-vector multiplication speed for various matrix sizes found in typical LLMs in Tab. 4. DBF is 2-3.5x faster with 2 bits/weights and 3-6x faster with 1 bit/weight.

For LLM decoding, we measure the time it takes to generate 128 tokens from an empty prompt, performed on compiled computational graphs, with batch size 1, and report the average number of generated tokens per second on a single GPU. We report results in Tab. 5. DBF achieves ~2.0-2.9x speedup compared to the fp16 baseline.

Although our empirical throughput numbers were timed on NVIDIA GPUs, DBF can be fast on a broad class of accelerators due to its flexibility and simplicity.

## 5  Conclusions and Future Work

In this paper, we introduced Double Binary Factorization (DBF), a novel compression method that approximates weight matrices as the product of two binary (sign) matrices, each accompanied by appropriate scaling vectors. The main benefit of DBF is replacing energy-costly multiplications with cheaper additions. DBF extends the concept of binary quantization, achieving competitive or superior compression performance compared to state-of-the-art quantization approaches. We demonstrated that DBF significantly outperforms existing methods using binary weight matrices and achieves comparable accuracy with state-of-the-art methods at compression rates of around 2 bits per weight. Furthermore, our approach is also practical, yielding speedups of 2-3.5x relative to dense baselines and promising significant energy savings due to the computational simplicity of binary operations, which only use additions instead of multiplications.

A distinct advantage of DBF is its inherent flexibility. Unlike many quantization approaches, DBF allows continuous control over compression ratios by adjusting the factorization's middle dimension. We further leveraged this flexibility by introducing an iterative, importance-driven pruning approach to dynamically allocate compression budgets across layers, enhancing overall compression efficacy.

There are multiple directions for future research to address DBF limitations. One of the main limitations of DBF is the problematic fine-tuning of binary matrices. While PV-tuning brings some benefits, we think that this can be further improved by doing factorization on-the-fly during fine-tuning. Furthermore, integrating the iterative pruning scheme with fine-tuning would also be beneficial and allow simplification of the compression process.

**Acknowledgments**

This research was supported by funding from the Slovak Research and Development Agency grant APVV-24-0045 and by grants 1/0140/25, and 1/0538/22 from the Slovak research grant agency VEGA. Part of the research results was obtained using the computational resources procured in the national project National competence centre for high performance computing (project code: 311070AKF2) funded by European Regional Development Fund, EU Structural Funds Informatization of society, Operational Program Integrated Infrastructure. We acknowledge the EuroHPC Joint Undertaking for awarding this project access to the EuroHPC supercomputer LEONARDO, hosted by CINECA (Italy) and the LEONARDO consortium through an EuroHPC Benchmark and AI Access calls.

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
