# OpenReview forum: "Addition is almost all you need: Compressing large language models with double binary factorization"
_TMLR — Accepted by TMLR_

### Review · Reviewer_krJk · 2025-12-10

**Summary Of Contributions:**

This paper proposes Double Binary Factorization (DBF), a novel matrix compression scheme that decomposes each weight matrix into a product of two binary sign matrices combined with learnable scaling vectors. By adjusting the size of the intermediate dimension, DBF provides continuous control over the effective compression rate, enabling fine-grained tuning of the average bits per weight—an advantage over conventional quantization methods that support only discrete bitwidth options. Empirically, DBF delivers competitive perplexity across a broad range of compression levels (1–2.3 bits/weight), outperforming prior 1-bit binarization approaches and achieving accuracy comparable to state-of-the-art quantizers at ~2 bits/weight, while enabling faster inference by replacing multiplications with additions. The authors further propose an importance-aware pruning mechanism for non-uniform per-layer compression.

**Audience:**

Yes

**Audience Explanation:**

In general, the deployment of large scale model is heavily dependent on the quantization methods.

**Claims And Evidence:**

Yes

**Claims Explanation:**

This paper has detailed experiment on wikitext to support the idea that their method is general enough to perform well on different degrees of compression. It can be adapted to different devices to balance the computation cost and the performance.

**Requested Changes:**

It is desirable to see the scaling property such as when we increase the hidden dimension / depth, what's the performance scaling property on some small-scale task or synthetic task.

---

> ### Author Response · Authors · 2026-01-04
>
> Thank you for your feedback.
>
> We investigated scaling of matrix sizes in Figure 3, where we compared how relative layer-wise loss changes when factorizing matrices from larger models.
> Unfortunately, doing full-scale 70B+ model compression (and especially fine-tuning) is outside our compute budget.

---

### Review · Reviewer_qhJ2 · 2025-12-22

**Summary Of Contributions:**

The paper introduces Double Binary Factorization (DBF), a post-training compression method that decomposes dense LLM weight matrices into a product of two binary sign matrices (±1) and three scaling vectors. The work is an extension to their prior ICLR 2025 submission (https://arxiv.org/pdf/2409.18850). By utilizing an ADMM-based heuristic to solve the non-convex factorization problem, the authors achieve competitive performance in the 1-2 bits per weight regime.

**Audience:**

Yes

**Audience Explanation:**

This work is of relevance to the journal's audience, particularly those focused on efficient machine learning and LLM deployment. As the scale of models grows, the transition from multiplication-heavy to addition-based inference is a significant research frontier. DBF offers a practical bridge between extreme binarization and high-precision quantization.

**Claims And Evidence:**

Yes

**Claims Explanation:**

The evidence for DBF’s effectiveness is clear and well-documented.

* Extreme Compression: DBF significantly outperforms OneBit and BiLLM in 1-bit settings.
* Speed: The method provides tangible 2-3.5x speedups on current hardware by replacing multiplications with additions.

**Requested Changes:**

My main comments are around two topics: contextual positioning & scalability.

**Contextual Positioning**: While the empirical results are strong, the paper’s discussion of related work is somewhat narrow. It misses several "joint factorization and quantization" approaches that provide necessary context:

1. Quantization Aware Factorization (Gusak et al., 2024): This is a significant omission. Like the current submission, it uses ADMM for joint factorization (specifically CP-decomposition) and quantization to reduce approximation error. A comparison or acknowledgement of how DBF’s binary constraint differs from their integer-grid constraint would be beneficial.

2. CALDERA (Saha et al., 2024): Recent work on approximating weights via $W \approx Q+LR$ (quantized backbone + low-rank factors) is highly relevant for the 2-bit regime where DBF claims competitiveness.

3. SVD-LLM & SVD-LLM V2 (Wang et al., 2025): The paper would benefit from discussing these truncation-aware SVD methods, which represent the current state-of-the-art for post-training factorization without binary constraints.

**Scalability**: While the results are strong, the paper acknowledges that at 4 bits and higher, standard scalar quantization still outperforms DBF. Additionally, the fine-tuning of binary matrices (PV-tuning) remains memory-intensive and complex. Further investigation into whether the higher-bit limitation is fundamental or algorithmic would strengthen the work.



Minor comments:

* Page 2: "...low-rank factorizations come with severe degradation in accura " -> accuracy.
* Page 3: "...we will select kask $k=b \frac{nm}{n+m}$"
* Page 5: "We use slight variation of the procedure...". It should be: "We use a slight variation...".
* Page 6: "...multiple state-to-art methods..." -> "state-of-the-art"
* Page 9: "...generate 128 tokens from emtpy prompt...". empty
​

---

> ### Author Response · Authors · 2026-01-04
>
> Thank you for the valuable feedback.
> We expanded the related work section with the works you suggested.
> We also added Caldera results to the experimental section.
>
> We also looked at the higher-bit limitation. It seems that altering the algorithm slightly (expanding the middle dimension over time) helps DBF for higher bit widths (3 bits and more). This suggests that the original DBF factorization algorithm was limited for higher bit widths.

---

### Review · Reviewer_G7Du · 2025-12-23

**Summary Of Contributions:**

Large Language Models demand significant memory and computation resources. Standard binary quantization reduces these costs but often severely degrades model accuracy. To address this, the authors propose Double Binary Factorization (DBF). This method factorizes dense weights into the product of two binary sign matrices and floating-point scaling vectors. This approach replaces expensive multiplications with additions and enables flexible compression ratios by adjusting the inner dimension size. The authors evaluate DBF on Llama2 and Llama3 models using language modeling and zero-shot reasoning tasks. DBF outperforms OneBit in 1-bit perplexity and remains competitive with state-of-the-art methods in the 2-bit regime while achieving inference speedups of 2-3.5x on standard GPUs.

**Audience:**

Yes

**Audience Explanation:**

Yes. The findings are likely to be of significant interest to several sub-communities within the TMLR audience, particularly those focused on efficient machine learning and Large Language Model (LLM) deployment.

**Broader Impact Concerns:**

I do not see any broader concerns.

**Claims And Evidence:**

Yes

**Claims Explanation:**

Yes, the claims are generally well-supported, though the scope of the evidence has some notable gaps:

- The claim of superior accuracy in extreme compression is convincingly supported by experiments showing DBF achieves lower perplexity (9.57) than the OneBit baseline (9.73) on Llama2-7B.

- Assertions regarding practical efficiency are substantiated by hardware benchmarks on an RTX 4090, which demonstrate clear 2-3.5x inference speedups over dense baselines by utilizing addition-only operations.

- The method's flexibility is accurately validated through ablation studies showing that adjusting the factorization's intermediate dimension allows for fine-grained, fractional bit-width targets (e.g., 1.5 bits) unlike standard integer quantization.

- However, the evidence is not fully comprehensive as the evaluation relies solely on perplexity and simple zero-shot tasks, omitting complex reasoning benchmarks (like GSM8K) or any computer vision tasks despite the broad title.

**Requested Changes:**

- Evaluation on Complex Benchmarks : The current evaluation relies primarily on Perplexity and simple zero-shot tasks (ARC, PiQA). To demonstrate that extreme quantization does not disproportionately harm complex reasoning capabilities, please evaluate the method on MMLU (Massive Multitask Language Understanding) and GSM8K (Grade School Math). These are standard "stress tests" for quantization that often reveal brittleness not captured by WikiText perplexity. (Required)

- Clarification of Scope or Title The paper is titled "Compressing neural networks...", yet all experiments are restricted exclusively to text-only Large Language Models (LLMs). Please address this discrepancy by doing one of the following:

- Modify the title to reflect the specific scope (e.g., "Compressing Large Language Models...").

- (Optional but encouraged) Include a preliminary experiment on a VLM (e.g., Llama 3.2 Vision or LLaVA). Since the method is already optimized for Llama, demonstrating it on a multimodal Llama variant would be a natural and compelling way to justify the broader "Neural Networks" claim without switching architectures entirely. Note: This experiment is not required for acceptance, but would significantly strengthen the paper's claims.

---

> ### Author Response · Authors · 2026-01-04
>
> Thank you for the feedback.
>
> Based on your suggestion, we added MMLU and GSM8k evaluation. Although evaluation on GSM8k is not typical in the LLM compression works we are familiar with (e.g., it is absent in QTIP, PV-tuning, QUIP# papers).
> On the other hand, our results show that all tested methods perform poorly on GSM8k.
> We do not yet know whether this is a problem with the fine-tuning data or a deeper limitation of compression methods.
>
> We also altered the title based on your suggestion.
> While we believe that our method would work on vision models, it is best suited for batch-size-1 decoding of LLMs, so we will stick with that.

---

### Review · Reviewer_7mr1 · 2026-02-24

**Summary Of Contributions:**

This paper focuses on binary quantization methods used in Large Language Model (LLM) compression. The authors propose a method called Double Binary Factorization (DBF) in order to achieve superior compression to Single Binary factorization and rates that claim to rival existing binarization/quantization approaches. The approach is flexible and allows for continuous control over compression ratios. The authors propose an algorithm for DBF and provide a numerical study comparing the compression and approximation errors with other state-of-the art methods.

**Audience:**

Yes

**Audience Explanation:**

I believe this work has broad interest to researchers in data compression and information theory. It would also appeal to researchers in theoretical machine learning, as the method serves as a benchmark for matrix (and tensor) compression techniques in LLMs.

**Broader Impact Concerns:**

I do not see any broader impact concerns.

**Claims And Evidence:**

Yes

**Claims Explanation:**

The authors largely claim that: 1) DBF can improve compression performance due to it's increased flexibility over other methods and 2) DBF may be performed faster (or take advantage of hardware specialized for accelerated computation) due to the binary matrices that require only additions. This work has numerical results that generally support both claims.

**Requested Changes:**

**My comments can be categorized into two major areas:**

1. General format and readability
a) This work focuses on a special category of neural networks called large language models and the authors frequently reference neural network terminology such as “layers”, “weight matrices” and “activation function” without properly defining them. A simple figure defining a general LLM with weight matrices and activation functions  would produce a clearer and more engaging introduction to a broader audience. It will also allow the reader to better understand Figure 1., as well as the differences of the state-of-the-art methods in the current literature.

2. Experimental results and discussion:

a) Table 1 and Table 2 indicate that DBF allows for considerable improvements in the 1-1.5 bit regime. In the 2-2.3 bit regime, the authors claim DBF to be comparable to AQLM and better than QUIP#. In fact, DBF also seems comparable to QUIP# and the improvement in performance seems marginal. What I think would help the authors and the quality of the work is if the work contained the answers to the following questions:
	*Under which application or regime would a practitioner be interested in 1 bit uniform compression? Why is such a compression useful?*

b) Concern with figure 2: the impact of DBF on approximation error is unclear. I agree that the figure shows that DBF has lower approximation error for weights with higher importance. However, with all methods, the approx error (irrespective of weight importance) is at most $2\times10^-3$ (and smoothed version is always less than $5\times10^-4$), which is already a low error.
	*How are you mathematically defining weight importance? How are you mathematically defining approximation error?*

c) I believe an expanded discussion on the enhanced computational speed/complexity of DBF compared to other methods is needed. The work claims this to be a benefit of DBF, where it can replace costly multiplications with cheaper additions. When considering the tradeoff between accuracy and speed, some practitioners/applications may tolerate slightly lower accuracy if computational costs are significantly reduced. One table quantifying DBF speed compared to PTQ methods would be sufficient and would strengthen the discussion for the experimental results in Table 1 and 2.

**Minor issues:**
1. QuIP# and QTIP , GPTQ, BiLLM stated throughout text but not defined.
2. Weight Binarization is not formally defined. BitNet and Onebit are referenced without being explicitly defined as weight binarization approaches.
3. Minor typo on page 4: “while ADMM provably...”

---

### Author Response · Authors · 2026-01-27
**Summary of changes**

We thank the reviewers for their constructive and insightful feedback.

Here is the summary of the changes we made:
- Change the title from "compressing neural networks" to "compressing large language models".
- Add evaluation on MMLU, GSM8k
- Reevaluated DBF for wider bitwidths
- Expanded related work section
- Evaluated DBF behaviour for larger matrices

Finally, we uploaded a revised version of the paper, with new text highlighted in blue, changed text highlighted in green, and deleted text highlighted in red.

---

### Decision · Action_Editor_D6AE · 2026-02-24

**Recommendation:** Accept as is

**Additional Comments:**

I recommend accepting the paper as is, because all reviewers concerns have been addressed in the revision.

**Audience:**

Yes

**Audience Explanation:**

There are strong needs and interests in deploying LLMs in the field, and this paper shows the potential to improve both the runtime and the memory footprint.

**Claims And Evidence:**

Yes

**Claims Explanation:**

This is a paper about a compression algorithm for reducing the size of large language models, with the benefit of improving runtime and memory footprint. The claims are largely centered around empirical success, and all reviewers are convinced and find the claims to be supported by the results.